# Effects of Low-Level Blast on Neurovascular Health and Cerebral Blood Flow: Current Findings and Future Opportunities in Neuroimaging

**DOI:** 10.3390/ijms25010642

**Published:** 2024-01-04

**Authors:** Madison O. Kilgore, W. Brad Hubbard

**Affiliations:** 1Spinal Cord and Brain Injury Research Center, University of Kentucky, Lexington, KY 40536, USA; madison.kilgore@uky.edu; 2Department of Physiology, University of Kentucky, Lexington, KY 40536, USA; 3Lexington Veterans’ Affairs Healthcare System, Lexington, KY 40502, USA

**Keywords:** neurovascular unit, blood–brain barrier, traumatic brain injury, blast overpressure, autoregulation, vasospasm

## Abstract

Low-level blast (LLB) exposure can lead to alterations in neurological health, cerebral vasculature, and cerebral blood flow (CBF). The development of cognitive issues and behavioral abnormalities after LLB, or subconcussive blast exposure, is insidious due to the lack of acute symptoms. One major hallmark of LLB exposure is the initiation of neurovascular damage followed by the development of neurovascular dysfunction. Preclinical studies of LLB exposure demonstrate impairment to cerebral vasculature and the blood–brain barrier (BBB) at both early and long-term stages following LLB. Neuroimaging techniques, such as arterial spin labeling (ASL) using magnetic resonance imaging (MRI), have been utilized in clinical investigations to understand brain perfusion and CBF changes in response to cumulative LLB exposure. In this review, we summarize neuroimaging techniques that can further our understanding of the underlying mechanisms of blast-related neurotrauma, specifically after LLB. Neuroimaging related to cerebrovascular function can contribute to improved diagnostic and therapeutic strategies for LLB. As these same imaging modalities can capture the effects of LLB exposure in animal models, neuroimaging can serve as a gap-bridging diagnostic tool that permits a more extensive exploration of potential relationships between blast-induced changes in CBF and neurovascular health. Future research directions are suggested, including investigating chronic LLB effects on cerebral perfusion, exploring mechanisms of dysautoregulation after LLB, and measuring cerebrovascular reactivity (CVR) in preclinical LLB models.

## 1. Introduction

Traumatic brain injury (TBI) is a long-standing concern in the field of military neurotrauma. Recent attention has been directed towards blast-induced TBI, driven by the prevalence of explosion-related injuries during Operation Enduring Freedom (OEF) and Operation Iraqi Freedom (OIF) and the increased use of improvised explosive devices in combat. Estimates suggest that 10 to 20% of Veterans returning from these conflicts have suffered from a blast-induced TBI [1,2]. 

The instances of mild traumatic brain injury (mTBI) in the US military exceed those of moderate-to-severe TBI, with blast exposure being the leading cause of mTBI [3,4,5,6]. mTBI is characterized by a physiological disruption of brain function, where loss of consciousness or alterations in mental status at the time of injury serve as primary diagnostic criteria [1]. However, assessing the neurological impact of low-level blast (LLB) exposure is challenging due to the lack of acute symptomology and, therefore, lack of initial diagnosis, hence the term subconcussive blast exposure [1,5]. For instance, 55% of Veterans who served in Iraq and Afghanistan sustained two or more blast exposures but still screened negative for TBI [7]. 

The biological effects of LLB overpressure are not well understood but can potentially affect the health of active duty service members and Veterans throughout their lives. Soldiers are routinely exposed to occupational LLB during training operations, wall breaching activity, and firing of specific weaponry [8]. Previous work has shown that repeated LLB can lead to transient symptomatology [9] and conditions that persist throughout a service member’s military career [10]. For example, high occupational risk of LLB not only correlates with diagnoses of mild to moderate TBI but also with an increased likelihood of experiencing symptoms similar to those experienced after TBI, such as memory loss [10]. These include cognitive issues, headaches, hearing problems, non-headache pain, sleep disturbances, and behavioral health conditions such as anxiety, drug and alcohol dependence, and post-traumatic stress disorder (PTSD) [10,11,12,13]. 

In a study by Reid et al. [6], mTBI-diagnosed servicemen with a more extensive history of blast exposure reported a greater number of post-concussion symptoms, including difficulties with sleeping, hearing, remembering, and concentrating, along with feeling irritable, tired, and anxious [11]. Likewise, service members with a history of wall breaching, or career breachers, with cumulative LLB exposure display signs of neurocognitive impairment [13] and a greater number and severity of post-concussion symptoms compared to those without repetitive LLB exposure [11,12]. These findings indicate that LLB may have a dose-dependent effect on neurological outcomes and that prior blast exposure renders the brain susceptible to subsequent blasts, placing service members exposed to repeated occupational LLB at significant risk of developing neurological symptoms, including PTSD. Substantial research supports a relationship between blast-induced TBI and PTSD-related symptoms [14,15,16], and LLB exposure particularly poses a serious concern because it is often not associated with losing consciousness. Thus, service members sustaining LLB may be more susceptible to developing PTSD, as PTSD occurs less frequently when there is memory loss of the traumatic event [1,17,18]. Exploring the repercussions of mild blast TBI (mbTBI) on PTSD is particularly important considering the economic burden for Veteran healthcare. Outpatient treatment expenses for Veterans with mbTBIs surpass those with non-blast-related TBIs [19], and healthcare costs are also 3.5 times higher for Veterans with PTSD [20]. 

Despite the known sequelae of repeated LLB exposure, questions still remain regarding the impact of LLB on the nervous system. The perplexities surrounding the neurological outcomes of blast exposure have led to an array of preclinical and clinical research. Neurovascular dysfunction is one major hallmark of blast exposure, irrespective of severity [21,22,23]. Consequently, this review discusses the impact of repeated LLB on neurovascular health and cerebral blood flow (CBF), as blast exposure has been shown to affect cerebral vasculature in animals and humans [7,24,25,26,27,28,29,30,31], and how neuroimaging can be used as a translational diagnostic tool to explore these outcomes. Given the importance of maintaining healthy cerebral vasculature and sufficient CBF for neurological function, it is plausible that these critical processes are compromised in service members and Veterans with a history of LLB exposure and contribute to the array of symptoms observed following blast exposure.

## 2. Neurovascular Dysfunction following Low-Level Blast Exposure 

As the field of blast-induced neurotrauma, blast TBI, and LLB progress, it is important to understand the alignment of clinical and preclinical findings. One area of investigation in the field is the examination of neurovascular, neurovascular unit (NVU), and blood–brain barrier (BBB) outcomes. The NVU describes cellular interactions with the blood vessels of the brain. The NVU includes vascular cells (including endothelial cells, pericytes, and vascular smooth muscle cells), glial cells (including astrocytes, microglia, and oligodendroglia), and neurons [32]. The cell-to-cell communication is vital for maintaining BBB integrity and regulating CBF [32,33]. While it is difficult to isolate the pathological effects of LLB in humans, studies focused on cumulative wall breaching and mbTBI have analyzed neuroimaging and serum biomarkers to better understand the mechanisms of blast-induced brain injury [13,29,34]. Simultaneously, numerous preclinical studies have been performed to study primary blast overpressure as the main experimental variable. 

For preclinical research, various methods of blast simulation are used to emulate blast overpressure exposure experienced by military personnel, including open-field explosives, shock guns, and air pressure generators [1,22,35,36]. Shock tubes or blast simulators use compressed gases in conjunction with diaphragms to produce a range of tunable and repeatable blast parameters to study varying injury severity. Several studies have been conducted to determine blast parameters that represent military-relevant LLB blast exposure. Rodents exposed to static peak overpressures ranging from 74 kPa (~11 psi) [37,38,39,40,41] to 124 kPa (~18 psi) [42] exhibit symptomology in line with characteristics of mTBI, including no loss of consciousness or observable gross pathology in the brain, lungs, and other organs. The range of blast overpressures for mild blast severity depends on a number of variables, including animal species, positive overpressure duration, and animal orientation to the blast.

Several studies have shown that repeated LLB influences the integrity of cerebral vasculature. For instance, Gama Sosa et al. [43] demonstrated that rats exposed to a blast overpressure of 10.8 psi once a day for three consecutive days exhibit cerebral vasculature disorganization (as analyzed with micro-CT imaging) up to 13 months post-repeated blast, along with other vascular pathologies, including dissociation of vasculature from the brain parenchyma, vascular remodeling, astrocytic degeneration, and neuroinflammation. Other studies found acute increases in BBB breakdown following LLB [40,41] and blast TBI [44,45,46,47,48]. When investigating BBB integrity in the chronic phase, a study by Rubovitch et al. [49] found that mice subjected to LLB via open-field explosive detonation show a delayed onset of increased BBB permeability one month after blast exposure. Additionally, in a study by Li et al. [50], a single, open-field LLB induces changes in the luminal diameter of cerebral microvessels, loss of pericytes, and detachment of astrocytic end-feet from the basement membrane of blood vessel endothelium at 30 days post-blast in mice. Further, LLB leads to other signs of BBB damage, including endothelial cell swellings at seven days post-blast, reduced intact tight junctions at 30 days post-blast, and elevated levels of neurofilament light chain (NfL) 3 months post-blast [50]. The elevation of NfL in human cerebral spinal fluid (CSF) inversely correlates with BBB integrity [51]. 

Astrocytes are pivotal in regulating CBF and maintaining BBB integrity via communication with endothelial cells through end-feet [32]. Astroglial scarring is described in postmortem examinations of military service members with acute and chronic blast exposure [52]. Repeated LLB-exposed animals show decreased expression of neurofilament proteins and glial fibrillary acidic protein (GFAP) in isolated brain vascular fractions, indicating astrocytic abnormalities, for up to 6 weeks following three consecutive days of exposure to a blast overpressure of 10.8 psi [53]. Moreover, a single LLB at an overpressure of 11 psi acutely decreases GFAP levels in the amygdala [40]. 

Contrary to these findings, Heyburn et al. [41] reported GFAP-reactive astrocytosis following repeated LLB at an overpressure of 10 psi. However, unlike the previous studies, this study measured GFAP levels following 14 consecutive days of blast exposures. Other investigations have also observed elevated GFAP levels following LLB exposure, specifically in the hippocampus [21,54,55,56,57]. These findings suggest that changes in astrogliosis following LLB may depend on the number of blast exposures and the specific brain regions under investigation. 

Based on blood biomarkers in military personnel with a history of blast exposure, it is clear that BBB damage is an important factor [58]. Chronic alterations in markers of BBB integrity have been observed among Veterans with mbTBI. In a study by Meabon et al. [58], blood plasma levels of vascular endothelial growth factor-A (VEGF-A), a factor contributing to BBB permeability, were elevated in Veterans who, on average, were 5.5 years removed from the last sustained mbTBI. Notably, changes in VEGF are particularly vulnerable to both single and repeated LLB exposure in preclinical models [41,59,60], indicating that damage to the NVU and, consequently, the BBB is an important prognostic and therapeutic target in service members exposed to repeated LLB. 

## 3. The Use of Neuroimaging in Blast-Related Neurotrauma Research

While we know that blast exposure can lead to neurovascular pathological outcomes, better diagnostic tools are necessary to fully understand the downstream neurological effects of LLB and their connection to symptomology. Neuroimaging is an important translational link and may provide an avenue for prognostication for long-term outcomes in Veterans with a history of blast exposure [61]. As new research emerges to address these questions, the challenge arises to bridge the gap between preclinical and clinical discoveries. Fortunately, the versatility of neuroimaging allows for assessment in both human subjects with a history of blast exposure and animal models of LLB. This versatility facilitates a multifaceted examination of the neurological consequences of LLB and also aids in closing the translational gap between preclinical and clinical research. Ultimately, these efforts can lead to discovering ways to improve health conditions for Veterans and active-duty service members.

When evaluating blast-induced neurological damage, modern imaging technology offers a practical approach to identifying changes in vascular pathology. Modalities like magnetic resonance imaging (MRI), computed tomography (CT), and positron emission tomography (PET) are available for assessing TBI-related outcomes in both animals and humans. Because MRI is capable of detecting subtle neurological outcomes in mTBI [62], MRI scans are a well-suited diagnostic technique for determining LLB-related damage. 

Several MRI techniques have detected neurological abnormalities following military blast-related neurotrauma. Diffusion tensor imaging (DTI) is an essential diagnostic tool to assess microstructural changes following blast exposure. DTI acquires information on axonal health by generating different parameters, including fractional anisotropy (FA) and radial diffusivity (RD), to determine the directionality of water diffusion along axons [62]. Several clinical studies have used DTI to assess the neurological outcomes in service members and Veterans who have experienced blast overpressure exposure. Compiled evidence from DTI MRI indicates that mbTBI [63,64,65,66,67,68] and occupational LLB exposure [29] disrupt axonal health. Additionally, DTI tractography, which maps white matter tracts in the brain, has been implemented in studies of white matter integrity in Veterans with prior blast exposure and has revealed a negative, dose-dependent association between the number of blast exposures and FA [69,70]. 

In a study by Trotter et al. [71], Veterans with a history of blast exposure not only showed a similar graded association via DTI but also demonstrated a more rapid decline in white matter integrity with age compared to unexposed individuals, indicating that cumulative blast exposure may contribute to an accelerated aging process. A recent study by Stone et al. [34] supports these findings, revealing that special operators with repeated LLB exposure show elevated PET-neuroinflammation, a characteristic of neurovascular dysfunction and contributor to the progression of neurodegenerative diseases such as Alzheimer’s disease (AD), Parkinson’s disease (PD), and amyotrophic lateral sclerosis (ALS) [72].

To specifically study the effects of blast overpressure on CBF, arterial spin labeling (ASL) is a highly suitable MRI technique. ASL allows for the quantification of cerebral perfusion by magnetically labeling arterial blood water and has been used to detect cerebral-perfusion-related pathologies such as cerebral ischemia and vasospasm [73]. ASL has been used in clinical studies investigating CBF changes associated with military-related TBI [24,26] and blast exposure [7]. ASL has also detected CBF alterations in civilian cases of mild and moderate-to-severe TBI [24,27,74,75,76,77,78,79]. Additionally, in a study by Clark et al. [26], combined ASL and DTI MRI data from Veterans with mild or moderate TBI indicated that reduced CBF is associated with poor white matter integrity and that this dynamic relationship may be responsible for adverse outcomes associated with TBI. Pseudocontinuous ASL (pCASL) is also a preferred approach for assessing CBF, as pCASL has a high labeling efficiency and signal-to-noise ratio [73,80]. It is clear that MRI techniques that monitor brain perfusion and CBF are critical to understanding the mechanisms and progression of cumulative LLB effects.

## 4. Clinical Evidence of Blast-Induced Changes in Cerebral Blood Flow

Research investigating the effects of altered CBF provides insight into the potential mechanisms by which LLB impairs neurovascular health. As previously mentioned, diminished CBF levels are correlated with impaired neuronal function [7,81]. This association is evident in studies of neurodegenerative diseases, in which reduced CBF is linked to cognitive decline [82,83,84], white matter hyperintensities [85], and PTSD [86,87,88]. 

In the clinical setting, incidences of insufficient CBF have been reported following brain injury. A majority of TBI patients present with hypotension and hypoxia [81], and in some cases of severe TBI, patients have exhibited hypoperfusion, hyperemia, vasospasm, dysautoregulation, and cerebral ischemia [89]. From a chronic standpoint, multiple clinical studies have demonstrated that alterations in CBF are persistent in TBI patients and Veterans with a history of mild to moderate TBI from 1 month to 8 years post-injury [24,26,27,28,75]. Moreover, studies have shown that mild to moderate TBI-induced alterations in CBF are linked with chronic neurological outcomes. Specifically, reduced CBF in the thalamus is associated with neurocognitive deficits [24], and reduced CBF in the cingulate cortex is associated with poorer white matter integrity [26].

However, very few clinical studies have investigated the effects of LLB exposure on CBF, especially in the chronic phase. In one clinical study by Stone et al. [29], both active and formerly active breachers showed a trending decrease in relative CBF, particularly in the caudate nucleus, compared to controls, as measured by gadolinium perfusion imaging. In another study by Sullivan et al. [7], active duty service members and Veterans with prior blast exposure exhibited perfusion increases in distinct brain regions, including the right middle/superior frontal, supramarginal, and middle/superior temporal gyri, along with the lateral occipital cortex, posterior and anterior cingulate cortex, insulae, and occipital poles, which were positively correlative with the number of lifetime blast exposures, indicating that blast exposure may have a dose-dependent effect on cerebral perfusion. 

Furthermore, this study also brings attention to the fact that certain brain regions may vary in susceptibility to perfusion changes that are blast-dose-dependent. For instance, service members and Veterans with higher numbers of prior blast exposure, 15–500 blasts, exhibited an increase in perfusion in the left anterior cingulate cortex compared to unexposed individuals (0 blasts) and in the left inferior frontal gyrus, frontal orbital cortex, left angular gyrus, and supramarginal gyrus compared to those exposed to lower numbers of prior blast exposure (1–14 blasts). These results suggest that LLB may lead to decreased brain perfusion before subsequent blast events (>15 blasts), leading to compensatory vascular remodeling or chronic vasodilation [21]. Critically, this study found that blast exposure was more indicative of brain perfusion changes compared to diagnosed TBIs, which drives the need to understand the neurovascular consequences of subconcussive blast.

Based on the scarcity of published findings, it is evident that more research is needed to determine the effects of LLB exposure on perfusion and CBF. Consistent diagnostic techniques should be adopted across various clinical and preclinical studies to comprehensively understand how LLB affects CBF. To highlight this point, the ASL sequence to measure regional brain perfusion is included in the latest protocol to investigate the long-term effects of repeated blast exposure [90]. Neurovascular, perfusion, and CBF changes are important indices to include in the neurological monitoring after LLB.

## 5. Preclinical Evidence of Low-Level Blast-Induced Changes in Cerebral Blood Flow

ASL has been used to detect changes in CBF in animal models of TBI. For instance, ASL has shown altered CBF as a consequence of experimental TBI [80,91,92,93] and hemorrhagic shock [91]. Very few preclinical studies exist on the specific effects of LLB on CBF, especially in the chronic phase following blast exposure. However, preclinical studies have demonstrated that higher [25,30,94] and lower levels [25] of blast overpressure impact CBF in the acute phase. Only one study, by Bir et al. [25], has used ASL to demonstrate that rats exposed to varying blast overpressures, simulated by a shock tube, exhibit reduced CBF in specific brain regions up to 72 h post-blast. Blast overpressures as low as 90 kPa demonstrated reduced CBF in the auditory cortex, medial dorsal cortex, and thalamus at 72 h post-blast exposure. Interestingly, ASL also revealed that all blast animals demonstrated a 0–27% decrease in perfusion to the hippocampus in comparison to sham animals, with this region exhibiting significantly reduced CBF at 117 kPa between 48 and 72 h post-blast and 193 kPa at 24 h post-blast [25]. These findings are consistent with previous studies that have demonstrated that blast overpressures as low as 135 kPa cause reduced CBF up to 2 h post-blast [30] and overpressures up to 180 kPa reduce CBF up to 5 days post-blast [94].

These few studies all concentrate on the acute phase following blast TBI, specifically LLB. Do these changes persist or even develop to represent long-term neurovascular changes that are potentially modifiable after LLB? Fundamentally, the precise relationship between neuronal activity and cerebral vasculature plays a critical role in the brain’s ability to autoregulate and maintain proper CBF. This regulatory mechanism is essential for meeting the high oxygen and metabolic energy demands necessary for the healthy functioning of the brain [95]. mTBI challenges these demands, and an initial cerebral response of altered CBF after injury [78,96] often leads to chronic CBF impairment [97,98,99]. In fact, studies have indicated that reduced CBF coincides with numerous neurodegenerative diseases, including AD, dementia, ALS, and PD [100]. Therefore, we hypothesize that early brain CBF changes and secondary neurovascular deficits can lead to chronic perfusion alterations after LLB (Figure 1).

Together, this evidence indicates that further exploration of the effects of blast-induced changes in CBF is needed. ASL is a promising technique to investigate alterations in neurovascular health and would also serve as a relevant diagnostic tool to expand into LLB research. These findings indicate that some brain regions may be more vulnerable to lower levels of blast, and using more neuroimaging could allow a better understanding of how LLB impacts particular brain regions long-term.

## 6. Cerebral Blood Flow as a Link between LLB and PTSD

As previously mentioned, service members with a high occupational risk of LLB are more vulnerable to developing PTSD [1,17,18]. Interestingly, imaging studies have shown that CBF in specific brain regions is altered in individuals at various stages of PTSD onset [86,87,88,101,102,103,104]. As such, monitoring of CBF/brain perfusion may represent a link between LLB and PTSD. Studies have shown that the onset of PTSD correlates with CBF alterations as early as three months after a traumatic event [104], and changes in CBF have been observed in Veterans who have experienced both military-conflict-related trauma [87,88,103] and early-life trauma [86]. 

Consequently, exploring the effects of chronic LLB exposure on PTSD-related outcomes is particularly imperative for Veterans, as PTSD symptoms can persist for decades and, in some cases, have a delayed onset, starting at a minimum of 6 months following the traumatic event [105,106]. Both active duty service members and Veterans with a history of repeated LLB exposure may be susceptible to acute and chronic onset of PTSD. However, using neuroimaging to assess CBF offers a potential method for identifying individuals more predisposed to developing PTSD symptoms, thereby allowing for earlier intervention. 

## 7. Cellular Underpinnings of Progressive Perfusion Deficits after LLB

The exact cellular mechanisms that lead to CBF changes after LLB remain elusive. While more research is needed to understand exactly how blast exposure affects cerebral perfusion, evidence suggests that blast overpressure is linked to changes in cerebrovascular function [7,107]. Examining the long-term ramifications of exposure to LLB is particularly significant due to its potential to contribute to chronic deficits in cerebral perfusion by accelerating aging-related mechanisms in cerebrovascular dysfunction, such as reductions in nitric oxide (NO) availability and neurovascular oxidative stress [108]. Notably, preclinical investigations have demonstrated that blast exposure triggers acute oxidative stress [21,44,60,109,110] and alters NO production [60,111,112], linking both to disruptions in BBB permeability [44,109,111,112,113]. Intriguingly, similar patterns of neurovascular dysfunction are apparent in neurodegenerative diseases, such as AD, dementia, and PD [100]. For instance, reduced CBF is evident in the early stages of all three diseases [114,115,116,117], while increased BBB permeability is characteristic of their later stages [118,119,120,121]. Indeed, LLB exposure leads to chronic vascular remodeling and may modulate age-related vascular changes [43,53]. Together, these findings suggest that there may be an overlap between the underpinnings of neurodegenerative diseases and the secondary consequences of blast exposure that propagate changes in cerebral vascular health and perfusion. Therefore, active duty service members and Veterans may be more susceptible to vascular issues related to brain aging, highlighting the crucial need for early disease prevention in those exposed to blasts.

## 8. Future Directions for Assessing Cerebrovascular Dysfunction

Cerebral vasospasm is a notable complication observed in military-related TBI and most commonly occurs as an outcome of blast-induced neurotrauma [31,122]. Although the exact mechanisms responsible for heightened vulnerability to vasospasm following blast events are incompletely understood, an in vitro model created by Alford et al. [123] indicated that it could be attributed to the hypercontraction of blood vessels in response to a vasoconstrictor immediately following blast exposure. Additionally, at 24 h post-blast, tissues subjected to more severe injuries displayed a reduction in contractile stress, indicating a phenotypic switch in vascular smooth muscle cells towards a synthetic, non-contractile phenotype. These findings suggest that blast exposure not only induces increased vasoconstriction but may also ultimately lead to vascular remodeling and chronic dysautoregulation response.

Considering this, cerebrovascular reactivity (CVR) is another potential assessment tool for evaluating vascular function following blast exposure. CVR serves as an indicator of the ability of cerebral vasculature to accommodate changes in CBF demand [124]. CVR assessment is frequently utilized in studies of sport-related concussion and mTBI [124,125,126,127,128], primarily via blood-oxygenation-level-dependent functional magnetic resonance imaging (BOLD MRI). These sports-related concussion studies have revealed that CVR is altered acutely following injury [125] and can persist up to 3 months post-concussion despite symptom recovery [128]. Furthermore, in a study by Churchill et al. [125], acute changes in CVR were still being attenuated up to a year following concussion, indicating that head trauma can lead to long-term changes in CVR and that CVR measurement allows for visualization of the recovery process following injury.

There is evidence that blast exposure can impact CVR. Blast overpressure induces alterations in CVR in ex vivo vessel preparations within a few hours post-blast [129] and persisting up to 28 days post-blast [31]. Additionally, because NO plays a role in CVR, it could also indicate vascular endothelial function in relation to blast-induced injury [128,130]. In future studies, it would be interesting to use a combined neuroimaging approach to assess the impact of blast exposure on CBF and CVR. For instance, ASL imaging allows for the direct measurement of CBF within brain tissue, while BOLD MRI utilizes the distinct magnetic properties of oxygenated and deoxygenated hemoglobin [131]. While BOLD MRI does offer an indirect neuroimaging technique for CBF assessment, it requires a standardized delivery magnitude and duration of vasoactive stimuli across subjects to provide accurate measures of CVR [131]. Therefore, the concurrent use of ASL and BOLD imaging could be an insightful tool, facilitating exploring outcomes associated with blast exposure on CBF and vascular function. Because of the possibility of utilizing these assessment tools in both preclinical and clinical studies of blast exposure, neuroimaging of CBF and vascular function has great potential in advancing our understanding of cerebral autoregulation in the context of blast-related injuries. 

## 9. Conclusions

Neurovascular pathology is a well-established outcome of blast-induced TBI. Military service members exposed to blast exhibit vascular dysfunction, including vasospasm, hypotension, and altered cerebral perfusion. While the cellular underpinnings of vascular dysfunction in military LLB exposure are not fully understood, clinical studies have shown, via neuroimaging, that alterations in CBF could underlie neurological symptoms, such as PTSD. Further investigation is particularly needed to determine the effects of repeated LLB exposure on CBF, especially as clinical studies have indicated a correlation between the number and severity of post-concussion symptoms and the extent of prior blast exposure.

Preclinical models of military-related blast have allowed for the investigation of pathology related to changes in CBF, including alterations in BBB permeability. Specifically, LLB exposure compromises several components of the NVU, including vascular smooth muscle cells, endothelial cells, astrocytes, pericytes, and tight junction proteins. Despite the limited preclinical research on LLB effects on CBF, evidence shows that blast exposure impairs neurovascular health, including alterations in acute cerebral perfusion and ongoing BBB permeability. Clinical reports highlight that blast exposure, including LLB, leads to cerebral perfusion alterations and dysautoregulation, although the directionality and regionality of these effects are inconclusive.

LLB is hypothesized to accelerate neurovascular-associated age-related processes, including cerebrovascular dysfunction and progression of neurodegenerative disease, in military service members with a history of repeated blast exposure, such as career breachers. The exploration of the chronic effects of blast exposure on CBF and neurovascular health is especially imperative given the growing population of Veterans exposed to blasts throughout their military careers. Therefore, an urgent need exists to better understand how blast exposure affects the NVU and to pinpoint the downstream neurovascular consequences of LLB exposure. Neuroimaging techniques offer a universal approach to disease detection that helps integrate clinical and preclinical investigations and allows for longitudinal measurements of neurovascular health, including CBF and cerebrovascular reactivity, following LLB.

## Figures and Tables

**Figure 1 ijms-25-00642-f001:**
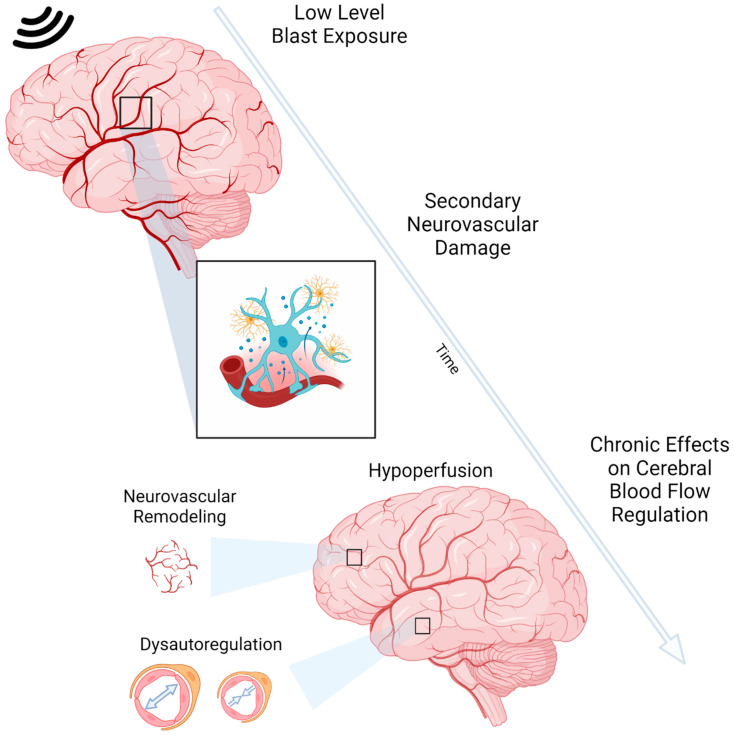
Development of neurovascular dysfunction over time following cumulative LLB exposure. Primary LLB exposure results in secondary neurovascular damage, including BBB breakdown, astrocytic alterations, inflammation, and pericyte loss. Ongoing neurovascular damage can lead to chronic effects on CBF and the regulation of cerebral perfusion, including hypoperfusion, neurovascular remodeling, and dysautoregulation. These consequences of cumulative LLB exposure contribute to neurological dysfunction and acceleration of brain aging mechanisms. Created with BioRender.com.

## Data Availability

No new data were created or analyzed in this study. Data sharing is not applicable to this article.

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
