# Peer review of "Effects of Low-Level Blast on Neurovascular Health and Cerebral Blood Flow: Current Findings and Future Opportunities in Neuroimaging"

_ijms, 2024, doi:10.3390/ijms25010642_

Round 1

Reviewer 1 Report

Comments and Suggestions for Authors

This concise review of low level blast neurotrauma is valuable contribution to research related to the effects of blast TBI, especially those related to cerebral blood flow and cerebral vascular function.  Aside from a few minor grammatical issues, the manuscript is clearly written and, in general, well organized.  However, there are several sections that would benefit from some revision.    

“In this review, we summarize neuroimaging techniques that can further our understanding of the underlying mechanisms of blast-related 19 neurotrauma, specifically after LLB.”   Since this review focuses on neuroimaging, some reference to imaging should be mentioned in the title.  

“LLB has also been shown to acutely decrease GFAP levels in the amygdala [37]. Other reports demonstrate GFAP-reactive astrocytosis following both repeated LLB and 126 blast TBI [19, 38, 51, 52].” (pg  3, lns 124 – 126)  These studies appear to be contradictory.  Since this is a review, the authors are encouraged to discuss potential reasons for the apparent contradiction.  For example, Hubbard, et al., (ref 37) found differences between male and female rats using a MacMillan shock tube.  Did the studies that observed GFAP-reactive astrocytosis use similar devices and/or injury levels, did all studies include male and female animals, etc.  

As noted by the authors, there are a variety of experimental models of blast neurotrauma.  In the review of pre-clinical literature, it would be helpful if the types of blast models (e.g. blast tubes vs shock tubes, type of shock tube, etc.) were noted.  

The information on blast effects on CBF is divided into pre-clinical and clinical subsections. For the sake of consistent organization, the authors might consider sub-divisions of other sections where appropriate.  

Comments on the Quality of English Language

The manuscript would benefit from minor editing for grammar.  For example, “Rubovitch et al. [46] found that mice subjected to LLB demonstrate a delayed onset of 111 increased BBB permeability one month after blast exposure” (pg 3, lns 110 – 112), used past and present tenses.  For the sake of consistency, the authors are encouraged to use a single tense in their discussions of previous research.  A related issue is that abbreviations (e.g. BBB, CBF) were used inconsistently.  

Author Response

We thank the reviewers for their time and effort dedicated to providing feedback on our manuscript. The insightful suggestions have greatly improved the clarity and cohesiveness of the review. We have incorporated most of the suggestions into our manuscript revision. Please see point-by-point responses in bold below each reviewer’s comments.

Reviewer 1

Comments and Suggestions for Authors

This concise review of low level blast neurotrauma is valuable contribution to research related to the effects of blast TBI, especially those related to cerebral blood flow and cerebral vascular function.  Aside from a few minor grammatical issues, the manuscript is clearly written and, in general, well organized.  However, there are several sections that would benefit from some revision.    

“In this review, we summarize neuroimaging techniques that can further our understanding of the underlying mechanisms of blast-related 19 neurotrauma, specifically after LLB.”   Since this review focuses on neuroimaging, some reference to imaging should be mentioned in the title.  

Thank you for suggesting a title adjustment. We have decided to change the title to “Effects of low-level blast on neurovascular health and cerebral blood flow: current findings and future opportunities in neuroimaging”

“LLB has also been shown to acutely decrease GFAP levels in the amygdala [37]. Other reports demonstrate GFAP-reactive astrocytosis following both repeated LLB and 126 blast TBI [19, 38, 51, 52].” (pg  3, lns 124 – 126)  These studies appear to be contradictory.  Since this is a review, the authors are encouraged to discuss potential reasons for the apparent contradiction.  For example, Hubbard, et al., (ref 37) found differences between male and female rats using a MacMillan shock tube.  Did the studies that observed GFAP-reactive astrocytosis use similar devices and/or injury levels, did all studies include male and female animals, etc.  

Thank you for this suggestion. We have expanded on these studies to discuss potential reasons for their differing results. We have included details for the studies that found opposing changes in GFAP levels, with a focus on contradictory findings being the result of 1) the number of blasts animals were subjected to 2) the specific brain regions that were examined (e.g. vascular fractions, amygdala, hippocampus).

As noted by the authors, there are a variety of experimental models of blast neurotrauma.  In the review of pre-clinical literature, it would be helpful if the types of blast models (e.g. blast tubes vs shock tubes, type of shock tube, etc.) were noted.  

Thank you. We have now incorporated specific text on studies that use open field blast simulation as well as limited references to shock tube studies in order to avoid redundancy. Please refer to both Section 2 and Section 5 for added text specifying the type of blast models.

The information on blast effects on CBF is divided into pre-clinical and clinical subsections. For the sake of consistent organization, the authors might consider sub-divisions of other sections where appropriate.  

Thank you for this suggestion on sub-dividing sections. While Section 2, “Neurovascular dysfunction following low-level blast exposure” provides both preclinical and clinical evidence of LLB-induced effects on the NVU, significantly less clinical research exists on the topic. There are not enough clinical studies on this topic to support its own subsection. By keeping a single section that includes preclinical and clinical results, we aim to provide a comprehensive summary of the correlations and disagreements between clinical and preclinical findings.

Comments on the Quality of English Language

The manuscript would benefit from minor editing for grammar.  For example, “Rubovitch et al. [46] found that mice subjected to LLB demonstrate a delayed onset of 111 increased BBB permeability one month after blast exposure” (pg 3, lns 110 – 112), used past and present tenses.  For the sake of consistency, the authors are encouraged to use a single tense in their discussions of previous research.  A related issue is that abbreviations (e.g. BBB, CBF) were used inconsistently.  

Thank you for locating this error in verb tenses and this has been fixed.

Thank you for locating inconsistencies with abbreviations. All subsequent references to “cerebral blood flow” have been changed to “CBF” after the initial use. We have also changed all subsequent references to the “blood brain barrier” to “BBB” after the initial use.

Reviewer 2 Report

Comments and Suggestions for Authors

It is well-known that exposure to low-level blast (LLB) can cause neurological alterations, cerebral vasculature, and cerebral blood flow (CBF). A major challenge is lack of acute symptoms, which can delay the treatment. Furthermore,

neurovascular damage followed by the development of neurovascular dysfunction are manifested.

cerebral vasculature impairment and damage to the blood-brain barrier at both early and long-term stages following LLB.

In this review, the authors provide an overview of neuroimaging techniques that can extend our understanding of the underlying mechanisms of blast-related 19 neurotrauma, specifically after LLB.

While this is an interesting review which summarizes the current understanding of LLB, there are several caveats which need to be addressed before the manuscript can be considered further.

1.     Title: please ensure it is specified in the title that this is a literature review

2.     Abstract Line 18: if this is the major objective of the manuscript, this should be reflected in the title: “In this review, we summarize neuroimaging techniques that can further our understanding of the underlying mechanisms of blast-related neurotrauma, specifically after LLB.”

3.     Abstract: “Future research directions are suggested, including investigating chronic LLB effects on cerebral perfusion, exploring mechanisms of dysautoregulation after LLB and measuring cerebrovascular reactivity (CVR) in preclinical LLB models” Given that is an important highlight of the manuscript, a separate section is needed titled “future directions” to discuss these directions

4.     Introduction: the first paragraph needs reorganization. It starts with TBI, then shifts to epidemiological data, and then followed by diagnostic and back again to epidemiology.

5.     Introduction: some definitions of mTBI should be provided to set the scene.

6.     Introduction: Line 48: this sentence is not clear and I don’t understand what authors mean: “For example, high occupational risk of LLB has not only been correlated with the diagnosis of mild to severe TBI but also with symptoms commonly comorbid with TBI”

7.     Introduction: Line 54: remove “Interestingly”

8.     Introduction: Line 68: I would argue this is not really the case and there is extensive research. “Despite the known sequelae of repeated LLB exposure, relatively little is known about the impact of LLB on central nervous system biology.”

9.      Introduction: Line 68: Also, it is not clear what is meant by “central nervous system biology” as this is such a broad term.

10.  The scope, aims and objectives of this manuscript should be mentioned at the end of introduction.

11.  Introduction: some data on economic burden of mTBI and LLB is required to strengthen the argument.

12.  Introduction should be enriched by mentioning some signs and symptoms of LLB.

13.  Introduction: Line 87: This sentence is not clear: “While it is difficult to isolate the clinical effects of LLB”

14.  Please ensure when each study is discussed, it is specified whether the study was on human or animals. For example, it is not clear from this sentence if this was a human or animal study: “Additionally, in a study by Li et al. [47], a single LLB induces changes in the luminal diameter of cerebral microvessels, loss of pericytes, and detachment of astrocytic end-feet from the basement membrane of blood vessel endothelium at 30 days post-blast”

15.  Line 129: Overall, the manuscript needs some editing, for example, BBB has been abbreviated before, but “blood-brain-barrier” is used here.

16.  The manuscript needs a table(s) summarizing major changes that occur to NVU and BBB

17.  Line 271: Discussion and future opportunities: Plenty of sections earlier actually belong to discussion. It is not clear why this section is the start of the discussion

18.  A section is needed at the end of the Discussion to summarize the limitations of the paper.

19.  Conclusion: some parts of the conclusion does not reflect the text. For example, it is mentioned in the conclusion that “LLB is hypothesized to accelerate neurovascular-associated age-related processes”, however, the only place where ‘age’ has been discussed is line 301 “Indeed, LLB exposure leads to chronic vascular remodeling and may modulate age-related vascular changes”

20.  Given that the population of veterans are aging, discussing the age-related consequences of LLB is of importance.

21.  While it is mentioned in the abstract that neuroimaging is discussed, this is just a small portion of the manuscript.

22.  Make sure p values are reported wherever relevant data from other studies are mentioned.

23.  “3. The use of neuroimaging in blast-related neurotrauma research” I suggest new methods, such as tractography and connectome are discussed to enrich the manuscript.

Comments on the Quality of English Language

Some minor editing is requored.

Author Response

We thank the reviewers for their time and effort dedicated to providing feedback on our manuscript. The insightful suggestions have greatly improved the clarity and cohesiveness of the review. We have incorporated most of the suggestions into our manuscript revision. Please see point-by-point responses in bold below each reviewer’s comments.

Reviewer 2

Comments and Suggestions for Authors

It is well-known that exposure to low-level blast (LLB) can cause neurological alterations, cerebral vasculature, and cerebral blood flow (CBF). A major challenge is lack of acute symptoms, which can delay the treatment. Furthermore, neurovascular damage followed by the development of neurovascular dysfunction are manifested. Cerebral vasculature impairment and damage to the blood-brain barrier at both early and long-term stages following LLB. In this review, the authors provide an overview of neuroimaging techniques that can extend our understanding of the underlying mechanisms of blast-related 19 neurotrauma, specifically after LLB.

While this is an interesting review which summarizes the current understanding of LLB, there are several caveats which need to be addressed before the manuscript can be considered further.

  1. Title: please ensure it is specified in the title that this is a literature review

Thank you for this suggestion. We feel the reader can ascertain that the paper is a review from the Abstract, Introduction, and the lack of primary data. However, we did adjust the title to incorporate neuroimaging (see suggestion #2).

  1. Abstract Line 18: if this is the major objective of the manuscript, this should be reflected in the title: “In this review, we summarize neuroimaging techniques that can further our understanding of the underlying mechanisms of blast-related neurotrauma, specifically after LLB.”

Thank you. The title has been changed to “Effects of low-level blast on neurovascular health and cerebral blood flow: current findings and future opportunities in neuroimaging”

  1. Abstract: “Future research directions are suggested, including investigating chronic LLB effects on cerebral perfusion, exploring mechanisms of dysautoregulation after LLB and measuring cerebrovascular reactivity (CVR) in preclinical LLB models” Given that is an important highlight of the manuscript, a separate section is needed titled “future directions” to discuss these directions

Thank you for this suggestion regarding a “future directions” section. Section 6 is now broken up into several separate sections, which includes “Future directions for assessing cerebrovascular dysfunction.”

  1. Introduction: the first paragraph needs reorganization. It starts with TBI, then shifts to epidemiological data, and then followed by diagnostic and back again to epidemiology.

Thank you for this suggestion on organization. We decided to keep the order of the sentences, since “For instance, 55% of Veterans who served in Iraq and Afghanistan sustained two or more blast exposures but still screened negative for TBI” supports the previous claim that the long-term neurological impacts of blast exposure fail to be diagnosed, especially as acute symptomology frequently does not meet criteria for a positive TBI screening. However, to improve the flow of the first paragraph of the introduction, we have separated the first paragraph into two separate paragraphs and added clinical manifestations of mTBI to address comment #5.

“Traumatic brain injury (TBI) is a long-standing concern in the field of military neurotrauma. Recent attention has been directed towards blast-induced TBI, driven by the prevalence of explosion-related injuries during Operation Enduring Freedom (OEF) and Operation Iraqi Freedom (OIF) and the increased use of improvised explosive devices in combat. Estimates suggest that 10 to 20% of Veterans returning from these conflicts have suffered from a blast-induced TBI [1, 2].

The instances of mild traumatic brain injury (mTBI) in the US military exceed those of moderate-to-severe TBI, with blast exposure being the leading cause of mTBI [3-6]. mTBI is characterized by a physiological disruption of brain function, where loss of consciousness or alterations in mental status at the time of injury serve as primary diagnostic criteria [1]. However, assessing the neurological impact of low-level blast (LLB) exposure is challenging due to the lack of acute symptomology and, therefore, lack of initial diagnosis, hence the term subconcussive blast exposure [1, 5]. For instance, 55% of Veterans who served in Iraq and Afghanistan sustained two or more blast exposures but still screened negative for TBI [7].”

  1. Introduction: some definitions of mTBI should be provided to set the scene.

Thank you for this suggestion. See comment #4 for the revision.

  1. Introduction: Line 48: this sentence is not clear and I don’t understand what authors mean: “For example, high occupational risk of LLB has not only been correlated with the diagnosis of mild to severe TBI but also with symptoms commonly comorbid with TBI”

Thank you – we agree that this sentence could be clearer. The initial statement was meant to explain that a high occupational risk of LLB is correlated with two separate outcomes: likelihood of sustaining a mild to moderate TBI and symptoms commonly associated with TBI. The sentence has been revised:

“For example, high occupational risk of LLB not only correlates with diagnoses of mild to moderate TBI but also with an increased likelihood of experiencing symptoms similar to those experienced after TBI, such as memory loss [10].”

  1. Introduction: Line 54: remove “Interestingly”

Thank you – this has been removed.

  1. Introduction: Line 68: I would argue this is not really the case and there is extensive research. “Despite the known sequelae of repeated LLB exposure, relatively little is known about the impact of LLB on central nervous system biology.”

Thank you for this suggestion. We agree that there is extensive research on the consequences of LLB exposure, but it’s impact on the central nervous system is incompletely understood. However, “relatively little is known” may be too drastic of an assumption. We have reworded Line 68 to say, “Despite the known sequelae of repeated LLB exposure, questions still remain regarding the impact of LLB on the nervous system.”

  1. Introduction: Line 68: Also, it is not clear what is meant by “central nervous system biology” as this is such a broad term.

Thank you – we have reworded Line 68 to say, “Despite the known sequelae of repeated LLB exposure, questions still remain regarding the impact of LLB on the nervous system.”

  1. The scope, aims and objectives of this manuscript should be mentioned at the end of introduction.

Thank you for this suggestion. The objective of this study is stated at the end of the introduction and we incorporated an additional objective of exploring outcomes of LLB exposure with neuroimaging:

“Consequently, this review discusses the impact of repeated LLB on neurovascular health and cerebral blood flow (CBF), as blast exposure has been shown to affect cerebral vasculature in animals and humans [7, 22-29], and how neuroimaging can be used as a translational diagnostic tool to explore these outcomes.”

  1. Introduction: some data on economic burden of mTBI and LLB is required to strengthen the argument.

Thank you. We have incorporated more information in the Introduction to strengthen the argument that the relationship between blast exposure and PTSD diagnoses is particularly imperative in the context of its economic burden on Veterans:

“Exploring the repercussions of blast-induced mTBI on PTSD is particularly important considering the economic burden for Veteran healthcare. Outpatient treatment expenses for Veterans with blast-related mTBI surpass those with non-blast-related TBIs [19] and healthcare costs are also 3.5 times higher for Veterans with PTSD [20].

  1. Introduction should be enriched by mentioning some signs and symptoms of LLB.

Thank you for this suggestion. We have incorporated specific symptoms to provide more detail on the post-concussive symptoms observed in the clinical studies that are mentioned in the introduction:

“In a study by Reid et al. [6], mTBI-diagnosed servicemen with a more extensive history of blast exposure reported a greater number of post-concussion symptoms including, including difficulties with sleeping, hearing, remembering, and concentrating along with feeling irritable, tired, and anxious [11]. Likewise, service members with a history of wall breaching, or career breachers, with cumulative LLB exposure display signs of neurocognitive impairment [13] and a greater number and severity of post-concussion symptoms compared to those without repetitive LLB exposure [11, 12].”

  1. Introduction: Line 87: This sentence is not clear: “While it is difficult to isolate the clinical effects of LLB”

Thank you – this has been reworded to say: “While it is difficult to isolate the pathological effects of LLB in humans, studies focused on cumulative wall breaching and mild blast TBI have analyzed neuroimaging and serum biomarkers to better understand the mechanisms of blast-induced brain injury [13, 27]. In conjunction, numerous preclinical studies have been performed to study primary blast overpressure as the main experimental variable.”

  1. Please ensure when each study is discussed, it is specified whether the study was on human or animals. For example, it is not clear from this sentence if this was a human or animal study: “Additionally, in a study by Li et al. [47], a single LLB induces changes in the luminal diameter of cerebral microvessels, loss of pericytes, and detachment of astrocytic end-feet from the basement membrane of blood vessel endothelium at 30 days post-blast”

Thank you – it has been added that this study was conducted on mice.

  1. Line 129: Overall, the manuscript needs some editing, for example, BBB has been abbreviated before, but “blood-brain-barrier” is used here.

Thank you for locating inconsistencies with abbreviations. All subsequent references to “blood brain barrier” have been changed to “BBB.”

  1. The manuscript needs a table(s) summarizing major changes that occur to NVU and BBB

This is an interesting suggestion but given the fact that the major emphasis on this paper is on CBF and neuroimaging, we feel that this is beyond the scope of this review. While other comprehensive reviews, such as the one referenced (https://doi.org/10.3389/fnins.2020.00581) have focused on TBI-induced damage on the NVU (vascular cells, glial cells, and neurons) and BBB, our objective is to specifically concentrate on changes in CBF using neuroimaging as the primary assessment tool.

In this review, we do reference blast-induced changes to the NVU and BBB; however, these studies serve a supplementary role, aimed at providing additional context and potential molecular underpinning for the observed changes in CBF. 

  1. Line 271: Discussion and future opportunities: Plenty of sections earlier actually belong to discussion. It is not clear why this section is the start of the discussion

Thank you for the suggestion and we agree with the reviewer’s assessment. We have renamed the “Discussion” section to be more specific to the idea(s) we expand upon in those paragraphs.

  1. A section is needed at the end of the Discussion to summarize the limitations of the paper.

Thank you for this suggestion regarding a limitations section. As limitation sections are not common for review articles, the only limitation we could propose would be that this was not conducted in a systematic review style. However, considering the paucity of articles on vascular neuroimaging in LLB, we did not think a systemic review was warranted. Since this manuscript is a review and not a systematic review, we have decided not to include a limitations section. A goal of this review is to point out that are broad limitations within LLB research. To address this, it is mentioned in the Conclusion that there is limited preclinical evidence on the effects of LLB on CBF and that further research is needed to understand the long-term effects of LLB.

  1. Conclusion: some parts of the conclusion does not reflect the text. For example, it is mentioned in the conclusion that “LLB is hypothesized to accelerate neurovascular-associated age-related processes”, however, the only place where ‘age’ has been discussed is line 301 “Indeed, LLB exposure leads to chronic vascular remodeling and may modulate age-related vascular changes”

We appreciate your suggestion on our concluding comments on LLB’s contribution to the age-related process. We have changed the conclusion sentence to better tie in the references we made to age-related processes and neurodegenerative disease progression. In Section 3 we elaborated on DTI studies that have found elevated neuroinflammation in blast-exposed individuals, which is an indicator of aging and a contributor to neurodegenerative disease progression (see comment #23). In Sections 5 and 6, we also discuss findings of similar alterations in CBF and BBB permeability between studies on blast-TBI and neurodegenerative diseases, such as AD, PD, dementia, and ALS.

  1. Given that the population of veterans are aging, discussing the age-related consequences of LLB is of importance.

Thank you. See comments #19 and #23 for adjustments.

  1. While it is mentioned in the abstract that neuroimaging is discussed, this is just a small portion of the manuscript.

Thank you for this suggestion. Section 3 is entirely dedicated to neuroimaging, and neuroimaging is discussed in Sections 4 and 5 pertaining to specific preclinical and clinical studies that examine changes in CBF following blast exposure. We have also expanded upon DTI techniques in Section 3 (see comment #23).

  1. Make sure p values are reported wherever relevant data from other studies are mentioned.

We appreciate your suggestion regarding the inclusion of p-values. However, since this is a comprehensive review of existing literature, our focus is on summarizing previous findings rather than conducting new statistical analyses. The aim of our review is to provide a broad overview of preexisting knowledge on the effects of LLB on neurovascular health and CBF in a succinct and cohesive manner. Consequently, we have decided to not include p-values for data from other studies.

  1. The use of neuroimaging in blast-related neurotrauma research” I suggest new methods, such as tractography and connectome are discussed to enrich the manuscript.

Thank you for this suggestion. Based on the reviewer’s suggestion, we have incorporated additional DTI studies that use tractography, as these studies show chronic effects of blast exposure on WM integrity in Veterans. Additionally, we have incorporated a study by Trotter et al. and Stone et al. that allowed us to tie in blast exposure and aging-related mechanisms to address comment #20.

“Several MRI techniques have detected neurological abnormalities following military blast-related neurotrauma. Diffusion tensor imaging (DTI) is an essential diagnostic tool to assess microstructural changes following blast exposure. DTI acquires information on axonal health by generating different parameters, including fractional anisotropy (FA) and radial diffusivity (RD), to determine the directionality of water diffusion along axons [57]. Several clinical studies have used DTI to assess the neurological outcomes in service members and Veterans who have experienced blast overpressure exposure. Compiled evidence from DTI MRI indicates that mild blast traumatic brain injury [58-63] and occupational LLB exposure [27] disrupt axonal health. Additionally, DTI tractography, which maps white matter tracts (WM) in the brain, has been implemented in studies of WM integrity in Veterans with prior blast exposure and has revealed a negative, dose-dependent association between the number of blast exposures and FA [65, 66].

In a study by Trotter et al. [67], Veterans with a history of blast exposure not only showed a similar graded association via DTI but also demonstrated a more rapid decline in WM integrity with age compared to unexposed individuals, indicating that cumulative blast exposure may contribute to an accelerated aging process. A recent study by Stone et al. [32] supports these findings, revealing that special operators with repeated LLB exposure show elevated PET-neuroinflammation, a characteristic of neurovascular dysfunction and contributor to the progression of neurodegenerative diseases such as Alzheimer’s disease, Parkinson’s disease, and amyotrophic lateral sclerosis [68].”